# Improvement of the Genome Editing Tools Based on 5FC/5FU Counter Selection in *Clostridium acetobutylicum*

**DOI:** 10.3390/microorganisms11112696

**Published:** 2023-11-03

**Authors:** Eglantine Boudignon, Céline Foulquier, Philippe Soucaille

**Affiliations:** 1Toulouse Biotechnology Institute (TBI), National Institute of Applied Sciences (INSA), Université de Toulouse, 135 Avenue de Rangueil, 31077 Toulouse cedex 4, France; eglantine.boudignon@insa-toulouse.fr (E.B.); celine.foulquier@insa-toulouse.fr (C.F.); 2Institut National de Recherche pour l’Agriculture, l’Alimentation et l’Environnement (INRAe), UMR 792, 24 chemin de Borde Rouge-Auzeville, 31326 Castanet-Tolosan, France; 3Centre National de la Recherche Scientifique (CNRS), UMR 5504, 16 Avenue Edouard Belin, 31055 Toulouse cedex 4, France; 4(BBSRC)/EPSRC Synthetic Biology Research Centre (SBRC), School of Life Sciences, The University of Nottingham, University Park, Nottingham NG7 2RD, UK

**Keywords:** *Clostridium acetobutylicum*, genome edition, 5FU, PyrR

## Abstract

Several genetic tools have been developed for genome engineering in *Clostridium acetobutylicum* utilizing 5-fluorouracil (5FU) or 5-fluorocytosine (5FC) resistance as a selection method. In our group, a method based on the integration, by single crossing over, of a suicide plasmid (pCat-*upp*) followed by selection for the second crossing over using a counter-selectable marker (the *upp* gene and 5FU resistance) was recently developed for genome editing in *C. acetobutylicum*. This method allows genome modification without leaving any marker or scar in a strain of *C. acetobutylicum* that is *∆upp*. Unfortunately, 5FU has strong mutagenic properties, inducing mutations in the strain’s genome. After numerous applications of the pCat-*upp*/5FU system for genome modification in *C. acetobutylicum*, the *CAB1060* mutant strain became entirely resistant to 5FU in the presence of the *upp* gene, resulting in failure when selecting on 5FU for the second crossing over. It was found that the potential repressor of the pyrimidine operon, PyrR, was mutated at position A115, leading to the 5FU resistance of the strain. To fix this problem, we created a corrective replicative plasmid expressing the *pyrR* gene, which was shown to restore the 5FU sensitivity of the strain. Furthermore, in order to avoid the occurrence of the problem observed with the *CAB1060* strain, a preventive suicide plasmid, pCat-*upp-pyrR**, was also developed, featuring the introduction of a synthetic codon-optimized *pyrR* gene, which was referred to as *pyrR** with low nucleotide sequence homology to *pyrR*. Finally, to minimize the mutagenic effect of 5FU, we also improved the pCat-*upp*/5FU system by reducing the concentration of 5FU from 1 mM to 5 µM using a defined synthetic medium. The optimized system/conditions were used to successfully replace the *ldh* gene by the *sadh-hydG* operon to convert acetone into isopropanol.

## 1. Introduction

In recent years, solventogenic Clostridia have garnered significant attention in the post-genomic era, primarily owing to the comprehensive sequencing and annotation of their genomes [1,2]. This wealth of genomic information has provided valuable insights into the metabolism of these industrially important strains, thereby catalyzing new approaches to genetic analysis, functional genomics, and metabolic engineering for the development of industrial strains geared toward biofuel and bulk chemical production.

To facilitate these endeavors, various reverse genetic tools have been devised for solventogenic Clostridia. These tools include markerless gene inactivation systems, employing methods such as homologous recombination with non-replicative [3,4,5] and replicative plasmids [6,7,8,9], as well as the insertion of group II introns [10,11,12]. For all homologous recombination-based methods involving two crossing-over, the use of a counterselection technique is imperative. This may include employing CRISPR-Cas9 [13,14,15,16] or a counter-selectable marker, which have been constructed using the codon-optimized *mazF* toxin gene from *Escherichia coli* (under the control of a lactose-inducible promoter) [7], the *pyrE* [8] gene (encoding an orotate phosphoribosyl transferase, leading to 5-fluoroorotate (5FOA) toxicity), the *upp* gene (encoding an uracil phosphoribosyl transferase, leading to 5-fluorouracil (5FU) toxicity) [5,9], or the *codA* gene [4,17] (encoding a cytosine deaminase that converts 5-fluorocytosine to 5FU, which is further transformed into a toxic compound by the product of the *upp* gene).

It is worth noting that while strategies relying on 5FC/5FU selection are highly effective, they should be employed cautiously. 5FU is a well-known anticancer drug recognized for its mutagenic properties in human cancers [18]. These mutagenic attributes have been demonstrated in various organisms, including *Caenorhabditis elegans*, *E. coli*, and *Mycobacterium tuberculosis* [19,20,21]. Moreover, the discharge of 5FU in the environment can impact aquatic organisms because of its mutagenic power [22].

While working on the metabolic engineering of *C. acetotutylicum*, we identified that the *upp*/5FU selection method could no longer be used to further engineer the *CAB1060* strain [23] as it became resistant to high concentrations of 5FU. In this study, we will demonstrate that this phenotype was due to a mutation in the *pyrR* gene encoding a potential repressor of the pyrimidine operon. To fix this problem, we created a corrective replicative plasmid expressing the *pyrR* gene, building upon the work of Bermejo et al. [24]. Furthermore, in order to avoid the occurrence of the problem observed with the *CAB1060* strain, a preventive suicide plasmid was also developed, inspired by Foulquier et al. [5], featuring the introduction of a synthetic codon-optimized *pyrR* gene, which is referred to as *pyrR** with low nucleotide sequence homology to *pyrR*. Additionally, a synthetic defined medium was optimized allowing a reduction in the concentration of 5FU required for the counterselection by a factor of 200 (from 1 mM to 5 µM), minimizing then the mutagenic effect of 5FU.

## 2. Materials and Methods

### 2.1. Bacterial Strains, Plasmids and Oligonucleotides

The bacterial strains and plasmids used in this study are referenced in Table 1. The oligonucleotides used for PCR amplification that were synthesized and provided by Eurogentec (Seraing, Belgium) are listed in Table 2.

### 2.2. Growth Conditions

*E. coli strains* were grown in Luria–Bertani (LB) medium. *C. acetobutylicum* strains were maintained as spores in synthetic medium (SM) at −20 °C as previously described or, for non-sporulating strains, directly in degassed and sterile serum bottles at −80 °C [25,26]. Spores were activated by heat shock at 80 °C for 15 min. Strains were grown under anaerobic conditions at 37 °C in Clostridial Growth Medium (CGM) supplemented each time with 30 gL^−1^ of glucose [27] or in CGM supplemented with 20 gL^−1^ MES hydrate (Sigma-Aldrich^®^, St. Louis, MI, USA), synthetic medium (SM) or in SM supplemented with 20 gL^−1^ MES hydrate or in Reinforced Clostridial Medium (RCM) (Millipore, Burlington, MA, USA). The pH of CGM was adjusted at 6.0 or 5.2 with hydrochloric acid. The pH of RCM was adjusted at 5.8 with hydrochloric acid. The SM used for *C. acetobutylicum* contained the following per liter of deionized water: glucose, 30 g; KH_2_PO_4_, 0.50 g; K_2_HPO_4_, 0.50 g; MgSO_4_.7H_2_O, 0.22 g; acetic acid, 2.3 mL; FeSO_4_.7H_2_O 10 mg; para amino benzoic acid, 8 mg; biotin, 0.08 mg. For *C. acetobutylicum* liquid cultures, SM was complemented with NiCl_2_, 3 mg; ZnCl_2_, 60 mg; and nitriloacetic acid, 0.2 g. The pH of the medium was adjusted to 6.0 with ammonia. For solid media preparation, 1.5% agar was added to liquid media. The media were supplemented as needed with the appropriate antibiotic at the following concentrations: for *C. acetobutylicum*, erythromycin (Ery) at 40 µg/mL, clarithromycin (Clari) at 40 µg/mL, and thiamphenicol (Tm) at 10 µg/mL; for *E. coli*, carbenicillin (Cb) at 100 µg/mL and chloramphenicol (Cm) at 30 µg/mL. Stocks of 5-fluorouracil (5FU) and uracil (Sigma-Aldrich^®^, St. Louis, MI, USA) were prepared at 0.1 M in dimethyl sulfoxide (DMSO) (Sigma-Aldrich^®^, St. Louis, MI, USA) and stored at −20 °C.

### 2.3. DNA Manipulation

Genomic DNA was extracted from *C. acetobutylicum* strains using GenEluteTM Bacterial Genomic DNA Kits (Sigma-Aldrich^®^, St. Louis, MI, USA). Plasmid DNA was extracted from *E. coli* using NucleoSpin^®^ Plasmid or NucleoBond^®^ Xtra Midi kits (Macherey-Nagel, Düren, Germany). Phusion DNA Polymerase (New England Biolabs (NEB, Ipswich, MA, USA)) was used to generate PCR products according to the supplier’s standard protocols. OneTaq^®^ 2X Master Mix with Standard Buffer (NEB, Ipswich, MA, USA) was used to screen colonies by PCR according to the supplier’s standard protocols. Restriction enzymes, antartic phosphatase, and T4 DNA ligase (NEB, Ipswich, MA, USA) were used according to the manufacturer’s instructions. DNA fragments were purified from agarose gel using a Zymoclean^TM^ Large Fragment DNA Recovery Kit (Zymo Research, Irvine, CA, USA). DNA PCR fragments were purified using NucleoSpin^®^ Gel and PCR Clean-up (Macherey-Nagel, Düren, Germany). Plasmid DNA and DNA PCR fragments were sequenced using the Sanger method (Eurofins Genomics, Ebersberg, Germany). PCR were conducted on a T100^TM^ Thermal Cycler (Bio-rad, Hercules, CA, USA). DNA recombinations were performed using the GeneArt^TM^ Seamless Cloning and Assembly Kit (Invitrogen, Thermofisher Scientific, Emeryville, CA, USA). Synthetic genes were synthesized by Geneart (Thermofisher Scientific, CA, USA).

### 2.4. Design of pyrR*

The nucleotide sequence of the *pyrR* gene (*CA_C2113*) was codon-optimized to create a synthetic *pyrR* gene, named *pyrR**, with low nucleotide sequence identity to the wild-type *pyrR* gene but in which substitutions would be as silent as possible and would not affect protein folding and function [28,29]. To do that, based on the codon usage table of *C. acetobutylicum* downloaded at https://gcua.schoedl.de/ (accessed on 7 December 2020), synonymous codon substitutions were introduced at all positions where it was possible, i.e., at all positions where the substitution did not replace a frequent codon by a rare one or conversely. The synthetic gene was synthesized by Geneart (Thermofisher Scientific, CA, USA). The sequence of the wild-type *pyrR* gene and the synthetic *pyrR** are described in Table 3.

### 2.5. Construction of pCat-upp-pyrR^mut^

This plasmid was constructed, based on the pCat-*upp* described by Foulquier et al. [5], by introducing the *pyrR^mut^* gene containing the mutation *g.344C>T* encoding the PyrR A115V protein found in our mutant strain. The mutant *pyrR* gene was PCR amplified with Phusion DNA polymerase using genomic DNA from *CAB1060* as the template and PSC 75 and PSC 76 primers containing *Bam*HI restriction sites. The PCR fragment and the pCat-*upp* were digested by *Bam*HI for one hour at 37 °C. The plasmid was dephosphorylated with antarctic phosphatase for 30 min at 37 °C. The PCR fragment and the plasmid were purified with NucleoSpin^®^ Gel (Macherey-Nagel, Düren, Germany) and a PCR Clean-up kit (Macherey-Nagel, Düren, Germany). The PCR fragment was cloned by ligation into the plasmid with T4 DNA ligase overnight at 16 °C to obtain pCat-*upp-pyrR^mut^*. The ligation was transformed in One shot^TM^ TOP 10 chemically competent *E. coli* following the manufacturers’ instructions (Thermofisher Scientific, CA, USA).

### 2.6. Construction of *pCat-upp-pyrR**

This plasmid was constructed from the pCat-*upp* described by Foulquier et al. [5] by introducing the synthetic *pyrR** gene under the control of the thiolase promoter. The entire pCat-*upp* plasmid was amplified with Phusion DNA Polymerase using PSC 51 and PSC 52 for linearization and was purified with NucleoSpin^®^ Gel (Macherey-Nagel, Düren, Germany) and a PCR Clean-up kit (Macherey-Nagel, Düren, Germany). The *pyrR** gene was amplified with Phusion DNA Polymerase using the synthetic *pyrR** gene as the template. A first PCR was performed with PSC 61 and PSC 62 primers to amplify the *pyrR** gene with its native RBS and downstream homology arms. The first PCR fragment was purified with NucleoSpin^®^ Gel (Macherey-Nagel, Düren, Germany) and a PCR Clean-up kit (Macherey-Nagel, Düren, Germany). A second PCR was performed on the first PCR product with PSC 72 and PSC 62 to introduce the upstream homology arm. The final PCR fragment of *pyrR** was purified from agarose gel using the Zymoclean^TM^ Large Fragment DNA Recovery kit (Zymo Research, Irvine, CA, USA) and cloned into the linearized pCat-*upp* plasmid by recombination using the GeneArt^TM^ Seamless Cloning and Assembly kit (Thermofisher Scientific, CA, USA). The plasmid was transformed in One shot^TM^ TOP 10 chemically competent *E. coli* following the manufacturers’ instructions (Thermofisher Scientific, CA, USA).

### 2.7. Construction of *pCat-upp-pyrR*-Δldh*

This plasmid was constructed based on the *pCat-upp-pyrR** (this study) and the pCat-*upp*-*Δldh* described by Nguyen et al. [23]. The *pCat-upp-pyrR** plasmid was linearized by digestion with the *Bam*HI restriction enzyme for one hour at 37 °C and dephosphorylated with antartic phosphatase for 30 min at 37 °C. The pCat-*upp*-*Δldh* plasmid was digested by *Bam*HI, and the fragment containing the *ldh* homology arms was purified from agarose gel using a Zymoclean^TM^ Large Fragment DNA Recovery kit (Zymo Research, Irvine, CA, USA). The two fragments were ligated using T4 DNA Ligase overnight at 16 °C. The plasmid was transformed in One shot^TM^ TOP 10 chemically competent *E. coli* following manufacturers’ instructions (Thermofisher Scientific, CA, USA).

### 2.8. Construction of *pCat-upp-pyrR*-Δldh::sadh-hydG*

This plasmid was constructed based on the pCat*-upp-pyrR*-Δldh* plasmid (this study) by introducing an operon composed of *sadh* and *hydG* genes (GenBank: AF157307.2), with their own RBS, under the control of the *ldh* promoter. The pCat-*upp-Δldh* was digested by *StuI* for one hour at 37 °C, dephosphorylated with antartic phosphatase for 30 min at 37 °C and purified with NucleoSpin^®^ Gel (Macherey-Nagel, Düren, Germany) and a PCR Clean-up kit (Macherey-Nagel, Düren, Germany). *sadh* and *hydG* were amplified with Phusion DNA Polymerase from a synthesized *sadh_hydG* gene as the template. A first PCR was performed with PSC 104 and PSC 105 to amplify *sadh_hydG* genes and introduce the *ldh* promoter region upstream of *sadh* and the *ldh* terminator downstream of *hydG*. After purification with NucleoSpin^®^ Gel (Macherey-Nagel, Düren, Germany) and a PCR Clean-up kit (Macherey-Nagel, Düren, Germany), the first PCR product was amplified with Phusion DNA Polymerase using PSC 106 and PSC 107 to introduce upstream and downstream homology arms to recombine with the pCat*-upp-pyrR*-Δldh* plasmid. The final PCR fragment was purified from agarose gel using a Zymoclean^TM^ Large Fragment DNA Recovery kit (Zymo Research, Irvine, CA, USA) and cloned into the pCat*-upp-pyrR*-Δldh* by recombination using the GeneArt^TM^ Seamless Cloning and Assembly Kit (Thermofisher Scientific, CA, USA). The plasmid was transformed in One shot^TM^ TOP 10 chemically competent *E. coli* following the manufacturers’ instructions (Thermofisher Scientific, CA, USA).

### 2.9. Construction of *pSOS95-pyrR*

This plasmid was constructed based on the pSOS95 plasmid described by Bermejo et al. [24] by introducing the native *pyrR* gene under the control of the thiolase promoter. The pSOS95 plasmid was digested by *Bam*HI and *Sfo*I and purified from agarose gel using the Zymoclean^TM^ Large Fragment DNA Recovery Kit (Zymo Research, Irvine, CA, USA). The *pyrR* gene was amplified with Phusion DNA Polymerase using the genomic DNA from the *C. acetobutylicum* strain *Δcac1502* as a template. The PSC 58 and PSC 46 used for this amplification introduced the *Bam*HI restriction site upstream of the *pyrR* gene RBS and the *Sfo*I restriction site downstream of the *pyrR* gene. The PCR fragment was digested by *Bam*HI and *Sfo*I and purified with NucleoSpin^®^ Gel (Macherey-Nagel, Düren, Germany) and PCR Clean-up kit (Macherey-Nagel, Düren, Germany). The PCR fragment was cloned in the pSOS95 plasmid by ligation using T4 DNA Ligase overnight at 16 °C. The plasmid was transformed in One shot^TM^ TOP 10 chemically competent *E. coli* following the manufacturers’ instructions (Thermofisher Scientific, CA, USA).

### 2.10. Transformation Protocol

The transformation of *C. acetobutylicum* was performed by electroporation according to the following protocol. From a culture of *C. acetobutylicum* in CGM at A_620_ between 1 and 2, a new serum bottle with 50 mL of CGM was inoculated at A_620_ of 0.1. When the culture reached A_620_ between 0.6 and 0.8, the culture was placed on ice for 30 min and transferred under an anaerobic chamber (Jacomex, Dagneux, France), where all the following manipulations were performed. The cells were harvested by centrifugation at 7000× *g* for 15 min (Centrifuge 5430, Eppendorf, Framingham, MA, USA) and washed in 10 mL of ice-cold electroporation buffer (EB) composed of 270 mM sucrose and 10 mM MES hydrate at pH 6.0. Then, the pellet was resuspended in 500 µL of EB, and cells were transferred into a sterile electrotransformation vessel (0.40 cm electrode gap × 1.00 cm) with 5–100 µg plasmid DNA. A 1.8 kV discharge was applied to the suspension from a 25 µF capacitor and a 400 Ω resistance in parallel using the Gene Pulser (Bio-Rad, Hercules, CA, USA). Cells were transferred directly to 10 mL of warm CGM and incubated for 6 h at 37 °C before plating on RCM supplemented with the required antibiotics.

### 2.11. Microbiological Enumeration on Solid Media

*C. acetobutylicum* was cultivated in CGM until reaching an A_620_ of 0.55 (Libra S11, Biochrom, Cambridge, UK). Subsequently, the culture was transferred to an anaerobic chamber, and 100 µL of various dilutions (10^−1^ to 10^−6^) of the culture was plated onto CGM MES or SM MES agar supplemented with the necessary antibiotics, ranging from 0 to 200 µM for 5FU and from 0 to 50 µM for uracil. Following incubation at 37 °C for a period of 1 to 4 days, the resulting colonies were counted.

### 2.12. 5FU Selection Protocol

*C. acetobutylicum* was cultivated in CGM until reaching an A_620_ of 0.55 (Libra S11, Biochrom, Cambridge, UK). The spreading protocol was the same as described in Section 2.11. After isolation, 50 colonies were picked and plated on a fresh plate with the same concentration of 5FU. The plates were incubated at 37 °C from 1 to 2 days. Once the colonies had grown, they were picked and patched onto plates with and without thiamphenicol to determine the percentage of double crossing-over events. Colonies showing a double crossover phenotype were screened by PCR to verify that genome editing occurred.

### 2.13. Locus Verification in C. acetobutylicum after Metabolic Engineering

After the genome edition of *C. acetobutylicum*, the different loci were checked by PCR amplification. In order to check for the insertion of point mutations, the genome was amplified by PCR with the primers of the Table 4, and the PCR fragment obtained was sent for sequencing.

### 2.14. Analytical Procedures

Viability percentages were calculated using the following formula:$$Viability=\frac{x}{50}*100$$
with *x* corresponding to the number of colonies growing after replicating on a fresh plate without antibiotics or 5FU.

The double-crossing over percentage was calculated through the Tm sensitivity using the following formula:$$Tm\;sensitivity=\frac{y}{x}*100$$
with *x* corresponding to the number of colonies growing after replicating on a plate without antibiotics and *y* corresponding to the number of colonies growing after replicate on a plate with antibiotics.

Culture growth was monitored by measuring optical density over time using a spectrophotometer at A_620_ (Libra S11, Biochrom, UK). For sample analysis, glucose, acetate, butyrate, acetone, isopropanol, ethanol, and butanol concentrations were measured using High-Performance Liquid Chromatography (HPLC) analysis (Agilent 1200 series, Les Ulis, France). Before injection, samples were centrifuged at 15,000× *g* for 5 min (Centrifuge 5424, Eppendorf, Framingham, MA, USA), and the supernatants were filtered through a 0.2 µm filter (Minisart^®^ RC 4, Sartorius, Epsom, UK). The separations were performed on a Bio-rad Aminex HPX-87H column (300 mm × 7.8 mm) (Bio-Rad, Hercules, CA, USA), and detection was achieved using either a refractive index measurement or a UV absorbance measurement (210 nm). The operating conditions were as follows: temperature, 14 °C; mobile phase, H_2_SO_4_ (0.5 mM); and flow rate, 0.5 mL/min. Excel 2019 was used for statistical analyses.

## 3. Results

### 3.1. Identification of the 5FU Resistance of the Strain

The strain *CAB1060*, as detailed by Nguyen et al. [23], was developed through the utilization of the *upp*/5FU counterselection method. This genome-editing method, initially described by Croux et al. [8] and originally employing a replicative plasmid, was subsequently adapted into a suicide plasmid format, as outlined by Foulquier et al. [5].

After several genome modifications and the use of 5FU as a counterselection marker, the strain became resistant to 5FU even at a concentration of 1 mM, and its entire genome was sequenced. Many random mutations were found, including one that particularly caught our attention: the mutation *g.344C>T* located in the *pyrR* gene that introduced a A115V mutation in the PyrR protein. PyrR is a potential repressor of the pyrimidine operon, and it has been shown in other organisms that mutations in the *pyrR* gene or a complete deletion of the *pyrR* gene can lead to 5FU resistance [20,21,30]. The hypothesis put forward was that the mutated protein no longer performed its regulatory function, and the pyrimidine operon was overexpressed. The overexpression of the pyrimidine operon could result in the overproduction of UMP, which protects bacteria from the toxic effects of 5FUMP. Based on these data, we hypothesized that the observed mutation could be responsible for 5FU resistance in the strain.

### 3.2. Evaluation of the 5FU Sensitivity of C. acetobutylicum Strain (Wild-Type pyrR Gene)

The viability of the *C. acetobutylicum ∆cac1502* strain was evaluated both on a rich medium (CGM MES) and on a synthetic medium (SM MES) in the presence of various concentrations of 5FU (Table 5). In the absence of 5FU, no significant differences could be observed between the two media. On the other hand, the 5FU sensitivity of the strain was much higher when spread on SM MES (Table 5) compared to rich media.

According to Singh et al. [21], exogenous uracil protects the bacteria from the toxicity of 5FU. This fact was validated in *Mycobacterium tuberculosis*, in which the supplementation of uracil at 15.6 µM protected bacteria up to 25 µM 5FU. Without uracil supplementation, *Mycobacterium tuberculosis* was sensitive to 3.12 µM of 5FU [21]. Based on of the literature, we have assumed that the yeast extract added to the rich medium contains between 25 and 50 µM of uracil [31]. Therefore, we tested the protective effect of uracil against 5FU in *C. acetobutylicum* by adding uracil to the synthetic medium containing 5 µM of 5FU (Table 6).

As expected, we observed a protective effect of uracil even at a very low concentration (5 µM), which is 5 to 10-fold lower than the expected concentration due to the addition of yeast extract. Based on these results, and to minimize the concentrations of 5FU used, all of the following experiments were carried out in a synthetic medium.

### 3.3. Construction of a C. acetobutylicum Strain with pyrR^mut^ and 5FU Resistance Validation

First, we tried to reproduce the occurrence of the mutation *g.344C>T* in the *pyrR* gene obtained in the 5FU-resistant *CAB1060* strain. To test for the occurrence of mutations, we plated 2 × 10^8^ cells of *C. acetobutylicum* strain *Δcac1502* on SM MES agar plates with high concentrations of 5FU (25–50 µM). The occurrence of mutations in the *pyrR* and *upp* genes was analyzed. No mutation in the *upp* was observed, but many mutations appeared at different positions in the *pyrR* gene (Table 7). However, the original mutation found in the *pyrR* gene in the 5FU-resistant *CAB1060* strain was not obtained. To ensure that the mutation *g.344C>T* was responsible for the 5FU resistance of the *CAB1060* strain, it was decided to introduce it into the *C. acetobutylicum* strain *Δcac1502ΔuppΔcac3535* using a pCat-*upp-pyrR^mut^*. After the transformation with pCat-*upp-pyrR^mut^* and the 5FU selection, the insertion of *g.344C>T* mutation in the *pyrR* gene was verified by sequencing. As the *C. acetobutylicum* strain *Δcac1502ΔuppΔcac3535pyrR^mut^* obtained was *Δupp*, it would be resistant to 5FU, so a *upp* gene was added into the suicide plasmid pCat *upp*-*Δldh.* In a strain with a wild-type *pyrR* gene, the *upp* gene contained in the suicide plasmid results in a strain sensitive to 5FU. However, as shown in Table 8, in the strain mutated in the *pyrR* gene (*pyrR^mut^* strain), most of the cells were resistant to high concentrations of 5FU. This result showed that the single *g.344C>T* mutation in the *pyrR* gene was sufficient to obtain a strain resistant to 5FU and confirmed our hypothesis concerning the 5FU resistance of the *CAB1060* strain.

### 3.4. Restoration of the 5FU Sensitivity in a Strain Mutated in the pyrR Gene

#### 3.4.1. Restoration of 5FU Sensitivity to 5FU Resistant Strains by Overexpressing the *pyrR* Gene on a Replicative Plasmid

This first method consists of using a replicative plasmid overexpressing a wild-type version of *pyrR* to overcome the problems of 5FU selection when a strain mutated in the *pyrR* gene already has a pCat-*upp* integrated. The viability of the *C. acetobutylicum pyrR^mut^* strain with the pCat-*upp-Δldh* integrated at the *ldh* locus and the pSOS95-*pyrR* replicative plasmid was tested in the presence of erythromycin for the maintenance of the replicative plasmid overexpressing *pyrR* and/or thiamphenicol for the maintenance of the pCAT-*upp*-*Δldh* suicide vector. Whereas, previously, in the presence of Tm, most of the cells were resistant to 5FU (Table 8), by just overexpressing a wild-type *pyrR* gene in the same strain, we restore its sensitivity to 5FU even at low concentrations (Table 9).

Indeed, by overexpressing the *pyrR* gene, a selection with 5FU at a concentration of 5 µM is sufficient to allow a high frequency (>85%) of double crossing-over in a *pyrR*^mut^ strain with an integrated pCat-*upp* in the genome. This frequency was even higher with a frequency over 95% at 10 µM 5FU (Table 10). These results showed that it was possible to reverse the 5FU resistance of a strain mutated in the *pyrR* gene with a pCat-*upp* integrated into its genome. The use of a replicative plasmid overexpressing a native *pyrR* gene allowed the excision of the suicide vector at very low 5FU concentrations.

#### 3.4.2. Restoration of 5FU Sensitivity to 5FU Resistant Strains by Overexpressing a Synthetic *pyrR** Gene on a Suicide Vector

The second method to overcome the problems of 5FU selection when a strain is mutated in the *pyrR* gene consists of introducing a wild-type version of the gene directly in the suicide plasmid. However, to avoid any possibility of recombination between the *pyrR* gene carried by the suicide plasmid and the *pyrR* gene located on the chromosome, a codon-optimized version of the *pyrR* gene (*pyrR**) has been designed and used to construct a *pCat-upp-pyrR**.

Viability of the *C. acetobutylicum pyrR^mut^* strain with a *pCat-upp-pyrR*-Δldh* integrated at the *ldh* locus was tested in the presence or absence of Tm for the maintenance of the suicide vector, and no difference in viability was observed (Table 11).

Whereas previously, in the strain with the pCat*-upp-Δldh*, most of the cells were resistant to 5FU in the presence of Tm (Table 8), with the same suicide plasmid but containing the *pyrR**, no colony were obtained (Table 11). These results validated the functionality of the synthetic *pyrR** gene and showed that a single copy of *pyrR** was sufficient to restore the sensitivity to 5FU of a resistant strain.

Indeed, by overexpressing the *pyrR** gene directly on the suicide vector, a selection with 5FU at a concentration of 5 µM is sufficient to allow a high frequency (>90%) of double crossing-over in a *pyrR*^mut^ strain. This frequency was even higher than 98% with 10 µM of 5FU (Table 12). Thus, overexpressing a single copy of *pyrR** was sufficient to restore the sensitivity to 5FU of a resistant *pyrR*^mut^ strain.

### 3.5. Preventive Use of pyrR*

After demonstrating the efficiency of overexpressing *pyrR* curatively in a 5FU resistant strain, we wondered if we could use the same method preventively in a wild-type *pyrR* strain to avoid the development of 5FU resistance. However, we first wanted to check that a second copy of *pyrR* did not result in a too high 5FU sensitivity. The viability of the *C. acetobutylicum* strain *Δcac1502ΔuppΔcac3535* with a *pCat-upp-pyrR*-Δldh* plasmid integrated at the *ldh* locus was assessed after 5FU selections at low concentrations (5 and 10 µM). The clones obtained on 5FU plates were then replicated onto fresh plates with and without Tm, and the results showed that viability was not affected: after 5FU selections at 5 µM and 10 µM of 5FU, 98% and 100%, respectively, of the picked colonies were viable (Table 13). In parallel, the frequency of double crossing over, evaluated through the sensitivity to Tm, was shown to reach 100% (Table 13). These results confirmed that overexpressing *pyrR* can be a method used both curatively and preventively.

### 3.6. Insertion of sadh and hydG from C. beijerinckii at ldh Locus

After validating the new *C. acetobutylicum* genome-editing method using pCat-*upp-pyrR**, we wanted to test this tool to both delete and replace genes in a single step. The goal was to use the pCat-*upp-pyrR*-Δldh::sadh-hydG* suicide plasmid to delete the *ldh* gene and replace it with an operon to produce isopropanol in the *C. acetobutylicum pyrR^mut^* strain. This operon is composed of *sadh* and *hydG* genes from *Clostridium beijerinckii NRRL B59* (Figure 1), which encode for a primary–secondary alcohol dehydrogenase (SADH) [32,33] and a putative electron transfer protein (HydG) [34], respectively. In Figure 2, the different stages involved in the integration of the “isopropanol operon” at the *ldh* locus are described. After integrating the suicide vector at the *ldh locus* and 5FU selection at 5 µM, Tm sensitive colonies were selected and screened by PCR using external primers. Primers were designed outside the homology arms to discriminate between wild-type revertants and mutants with the desired genotype (*Δcac1502ΔuppΔcac3535 pyrR^mut^ Δldh::sadh hydG*) (Figure 3). After picking and patching colonies a second time on 5FU, we obtained 92% of viable colonies using 5FU at 5 µM. After selecting on Tm, all viable clones were shown to be sensitive to 5FU; i.e., 100% excision of pCat-*upp-pyrR** was achieved (Table 14). Both wild-type revertants and mutants with the desired genotype were obtained. In Figure 3, an example of two mutant clones (11 and 12) and one wild-type revertant clone (clone 15) is shown. The pCat-*upp-pyrR** tool can, therefore, be used to delete genes and replace them with others in a single step.

### 3.7. Culture of C. acetobutylicum on Synthetic Medium for Isopropanol Production

To validate the functionality of the “isopropanol operon”, the *C. acetobutylicum pyrR^mut^* strain *∆ldh::sadh-hydG* and the control strain, without isopropanol production pathway, were cultured in synthetic media in serum bottles at an initial pH of 6.0. After 48 h of culture, the production of solvents and acids, as well as the final product yields, were evaluated. Both strains had a growth rate of 0.13 h^−1^ during the first 15 h of cultures and reached an A_620_ maximum between 2.3 and 2.5 and entered the lysis phase after 15 h of cultures. As expected, isopropanol production was only detected in the *C. acetobutylicum pyrR^mut^* strain *∆ldh::sadh-hydG* with a final molar yield shown to be associated with a decrease in acetone production in comparison to the control strain (acetone production is 1.2-fold higher in the control strain) (Figure 4d). This strain also had lower acetate consumption, lower butyrate production, and slightly higher ethanol production (Figure 4d). The functionality of the “isopropanol operon” inserted at the *ldh* locus has thus been confirmed.

## 4. Discussion

In the present study, we have shown that the overexposure of *C. acetobutylicum* to 5FU can make it resistant to this drug. Spontaneous mutations are induced in the bacterial chromosome, notably in the *pyrR* gene. As described in other publications, the *pyrR* gene encodes for PyrR, which is the repressor of the pyrimidine operon [20,21,30]. The presence of a mutation in this protein can lead to the cessation of its function, resulting in an overproduction of UMP. This overproduction of UMP can protect against the harmful effects of 5FUMP, which is a molecule that is toxic to bacteria. We observed the appearance of a mutation in the PyrR protein of *C. acetobutylicum* that completely avoids the selection of the double crossing-over step when using the pCat*upp*/5FU system. The principle of the *upp*/5FU system was based on the use of a strain in which the *upp* gene has been deleted and the use of 5FU as a counterselection agent. The *upp* gene encodes for an uracil phosphoribosyl transferase (UPRTase) that could convert uracil to UMP and 5FU to 5FUMP. 5FUMP is a molecule that prevents cells from producing intermediates needed for DNA synthesis, thereby causing cell death (Figure 5a) [21]. According to Peters et al., 5FUMP and UMP compete for binding to thymidylate synthase. 5FUMP targets thymidylate synthase and inhibits its activity, which is used to convert UMP to TMP. The production of TMP is decreased, and subsequently, DNA production is decreased [35].

When the A115V PyrR mutation was discovered in *C. acetobutylicum*, we compared it with other mutations in homologous proteins already described in the literature. A conserved protein sequence required in PRPP binding can be found in many species. Ghode and Singh described the G125V and R126C mutations *in M. tuberculosis* as being in this conserved zone [20,21]. According to Ghode, a mutation in this region could block the production of 5FUMP. As the A115V PyrR mutation is situated close to this site, it could be one of the reasons why our strain is resistant to 5FU (Figure 6). In addition, *pyrR* encodes the regulatory protein of the pyrimidine operon. According to Ghode and Fields [20,36], after mutations in PyrR of *M. tuberculosis* or *M. smegmatis*, or when PyrR is completely deleted in *B. subtilis*, the protein no longer performs its regulatory function, and the pyrimidine operon is overexpressed [30]. This results in the overproduction of UMP, which protects the bacteria from the toxic effects of 5FUMP. This mechanism described in M*ycobacteria* seems to work in the same way in *C. acetobutylicum* as a single mutation occurrence in PyrR (A115V) caused a 5FU resistance of the strain. In a *C. acetobutylicum pyrR^mut^* strain, the exact mechanism (affected PRPP binding or impaired regulatory function with a consequent pyrimidine operon overexpression or both) leading to 5FU resistance is not known yet. The two hypotheses have been summarized in Figure 5b.

To overcome this problem, we had to revise the protocol previously described by our team. First, we realized that the composition of the media plays an essential role in the resistance of the strain to 5FU. It is preferable to use a synthetic media that is not supplemented with uracil. The yeast extract present in CGM brings uracil into the media and thus protects against the toxic effect of 5FUMP. This hypothesis was tested by adding low concentrations of uracil to the synthetic media. Bacterial growth was no longer affected by the presence of 5FU in the media. After optimizing the media for 5FU selection, we constructed two plasmids to restore the sensitivity of the strain to 5FU. The first plasmid is a replicative plasmid overexpressing a native version of *pyrR*. It is used to overcome the problems of 5FU selection when a strain mutated in the *pyrR* gene has already integrated a pCat-*upp*. The second plasmid is a pCat-*upp* containing a codon-optimized version of the *pyrR* gene called *pyrR**. The 5FU selection problem for genome editing is directly bypassed by this method. With both of these strategies, the concentration of 5FU could be reduced from 1 mM to 5 µM, thus minimizing the risk of spontaneous mutation. Both the use of *pyrR** and the use of the SM media will be beneficial for all the counterselection methods involving *upp* and 5FU as well as those utilizing *codA* and 5FC [4].

Once the new protocol was established, we demonstrated that it was possible to both delete and insert genes of interest in the *C. acetobutylicum ∆cac1502∆upp∆cac3535 pyrR^mut^* strain in a single step using the pCat-*upp-pyrR**/5FU system. An isopropanol production pathway from *C. beijerinckii* was inserted at the *ldh* of *C. acetobutylicum ∆cac1502∆upp∆cac3535 pyrR^mut^∆ldh::sadh-hydG* strain utilizing this technique. We decided to insert the *sadh* and *hydG* genes from *C. beijerinckii NRRL 593* following a publication by Dusséaux et al. [32]. SADH is an NADPH-dependent primary-secondary alcohol dehydrogenase that catalyzes acetone reduction, and HydG is a putative electron transfer protein [34,39]. *hydG* was introduced into the *C. acetobutylicum ∆cac1502∆upp∆cac3535 pyrR^mut^ strain* genome at the same time as *sadh*, since these two genes are located in the same operon in *C. beijerinkcii NRRL 593*. It was assumed that the HydG activity would positively affect the SADH activity, allowing the strain to obtain better isopropanol production [34]. The final production of our *C. acetobutylicum ∆cac1502∆upp∆cac3535 pyrR^mut^ ∆ldh::sadh-hydG* strain is lower than the one obtained by Dusséaux (up to 4.7 g·L^−1^ of isopropanol produced in a culture of 30 h) with a lower molar ratio of isopropanol/acetone [32]. This result can be explained by the fact that in this strain, both genes were overexpressed in a multi-copy replicative plasmid and under the control of the *ptb* promoter, which is a stronger promoter than the *ldh* promoter [40].

## 5. Conclusions

In this work, we provide an explanation of the 5FU resistance of the *CAB1060* mutant of *C. acetobutylicum*. Furthermore, we provide two tools: (1) one to fix the problem of the 5FU resistance of *C. acetobutylicum* to allow further engineering of the strain using 5FU/5FC resistance in the form of a replicative plasmid carrying the *pyrR* gene, and (2) one to prevent the phenomena of 5FU/5FC resistance by using a new suicide vector carrying both the *upp* gene and synthetic *pyrR** genes associated to a medium that allows the use of a low concentration of 5FU/5FC to minimize their mutagenic effect. Finally, the optimized system/conditions were used to successfully replace the *ldh* gene by the *sadh-hydG* operon to convert acetone into isopropanol. We hope that these tools will be helpful for the scientific community working on the genome editing and the metabolic engineering of *C. acetobutylicum*.

## Figures and Tables

**Figure 1 microorganisms-11-02696-f001:**
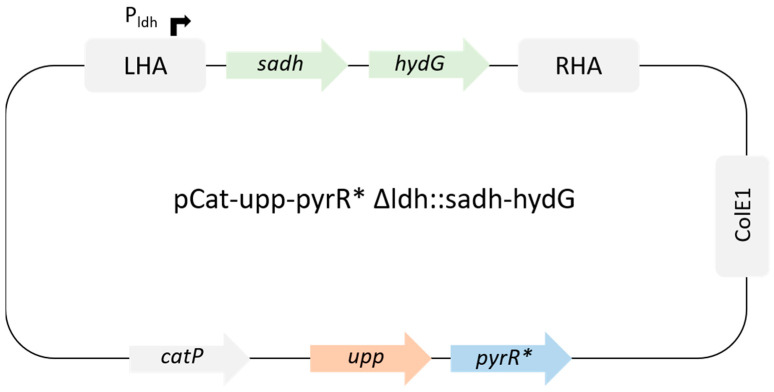
Suicide plasmid for *ldh* replacement by *sadh* and *hydG* from *C. beijerinckii*.

**Figure 2 microorganisms-11-02696-f002:**
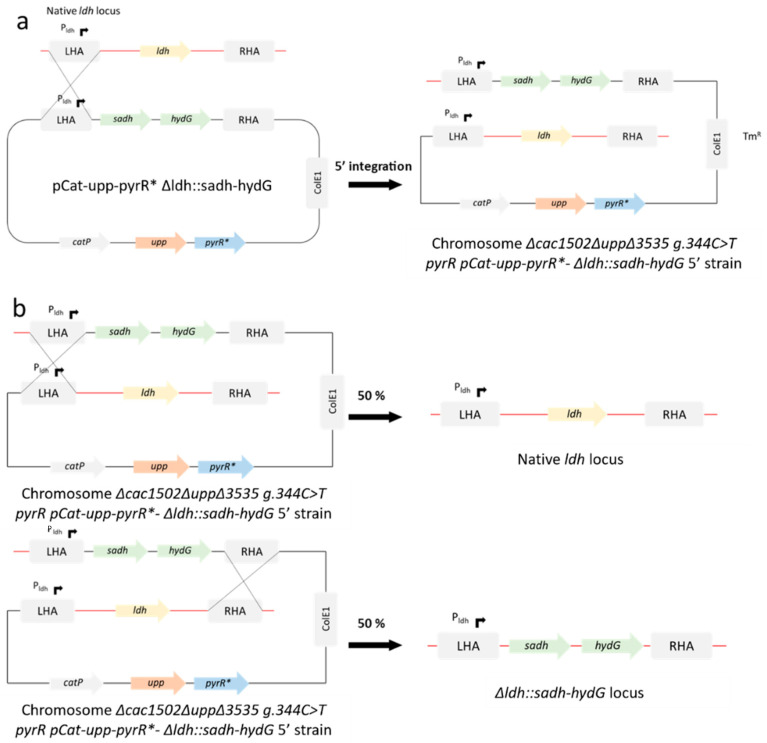
Diagram representing the replacement of *ldh* by *sadh and hydG* from *C. beijerinckii* by allelic exchange in a *pyrR^mut^* strain. LHA: left homology arm; RHA: right homology arm. (**a**) 5′ integration of the suicide plasmid. The integrants are selected on thiamphenicol. (**b**) Double crossing over induced by 5FU that causes the excision of the suicide plasmid.

**Figure 3 microorganisms-11-02696-f003:**
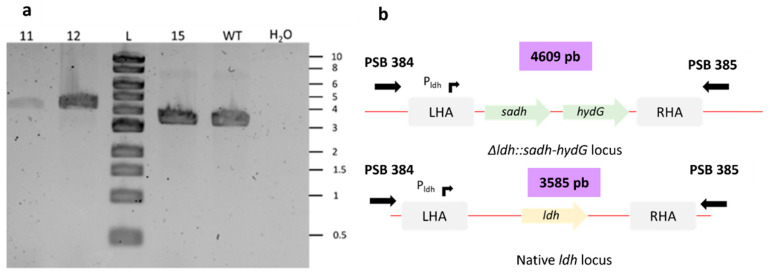
(**a**) Screening of *Δldh::sadh hydG* mutants. The colonies were screened using PSB 384 and PSB 385 primers. Ladder: 1 kb DNA ladder provided by New England Biolabs. (**b**) Schematic representation of *Δldh::sadh hydG* locus and native *ldh locus.*

**Figure 4 microorganisms-11-02696-f004:**
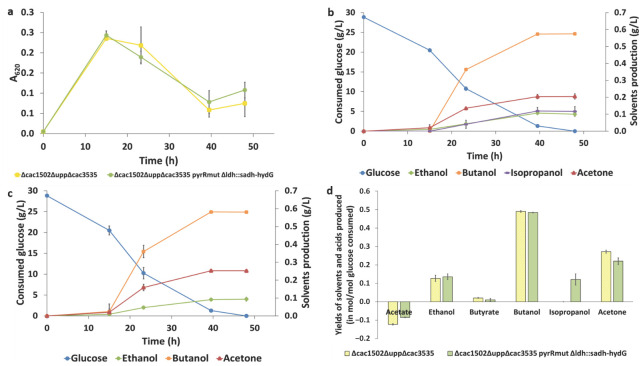
(**a**) Growth profile of the *C. acetobutylicum pyrR^mut^* strain *∆ldh::sadh*-*hydG* and *C. acetobutylicum* strain *∆cac1502∆upp∆cac3535* on SM. (**b**) Monitoring glucose consumption and solvents production over time of *C. acetobutylicum pyrR^mut^* strain *∆ldh::sadh*-*hydG.* (**c**) Monitoring glucose consumption and solvents production over time of *C. acetobutylicum* strain *∆cac1502∆upp∆cac3535.* (**d**) Molar yields of solvents production. All the measurements shown are mean average (*n* = 3). Errors bars represent the standard deviation.

**Figure 5 microorganisms-11-02696-f005:**
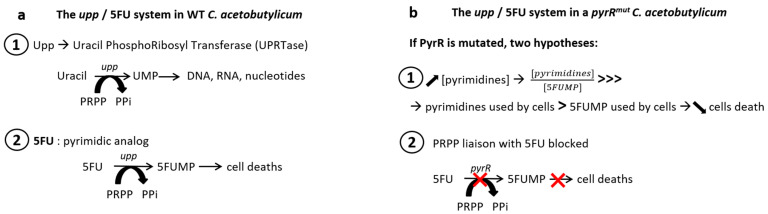
Metabolism of 5FU. (**a**) Metabolism of 5FU in a wild-type *C. acetobutylicum.* (**b**) Hypothetical effects of PyrR mutation in *C. acetobutylicum*. PRPP, phosphoribosyl pyrophosphatase; PPi, pyrophosphatase; UMP, uridine monophosphate; 5FU, 5-fluorouracile; 5FUMP, 5-fluorouridine monophosphate.

**Figure 6 microorganisms-11-02696-f006:**
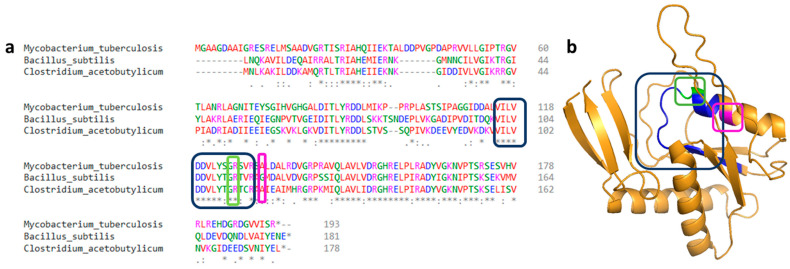
(**a**) Multiple sequences alignment of PyrR. (**b**) Cartoon diagram of PyrR protein of *C. acetobutylicum* predicted by Alpha-fold [37,38]. The blue boxes show the amino acids implied in PRPP binding. The green boxes highlight G125 and R126 sites described by Ghode and Singh [20,21]. The pink boxes represent the A115 position where *C. acetobutylicum* was mutated. * represents identical amino acids, ‘:’ represents amino acids that are strongly similar, ‘.’ represents amino acids that are weakly similar.

**Table 1 microorganisms-11-02696-t001:** Bacterial strains and plasmids used in this study.

**Strain or Plasmid**	**Relevant Characteristics**	**Source or Reference**
Bacterial strains		
*E. coli*		
TOP10		Invitrogen
*C. acetobutylicum*		
*CAB1060*	*Δ*CAC1502*ΔuppΔptbΔbukΔ*ctfAB*ΔldhAΔrexA ΔthlA::atoB Δhbd::hbd1*	[23]
*Δcac1502*	*ΔCA_C1502*	[9]
*Δcac1502ΔuppΔcac3535*	*ΔCA_C1502 ΔCA_C2879 ΔCA_3535*	[9]
*Δcac1502ΔuppΔcac3535 pyrR^mut^*	*Δ* *CA_C1502 Δ* *CA_C2879 Δ* *CA_3535 CA_C2113 g.344C>T*	This study
*Δcac1502ΔuppΔcac3535 pyrR^mut^ Δldh::sadh hydG*	*ΔCA_C1502 ΔCA_C2879 ΔCA_3535 CA_C2113 g.344C>T ΔCA_C0227:: CIBE_3470 HydG* (Accession: P25981.3)	This study
Plasmid		
pCat-*upp*		[5]
pCat-*upp*- *pyrR^mut^*		This study
	*Cm^R^*, *upp*, *colE1 origin*	
	*Cm^R^*, *upp*, *pyrR edition cassette for C. acetobutylicum*	
pCat-*upp-Δldh*		[23]
	*Cm^R^*, *upp*, *ldh deletion cassette for C. acetobutylicum*	
pCat-*upp-pyrR**	*Cm* *^R^ upp pyrR**	This study
pCat-*upp-pyrR*- Δldh*	*Cm^R^*, *upp pyrR**, *ldh deletion cassette for C. acetobutylicum*	This study
pCat-*upp-pyrR*-∆ldh::sadh-hydG*	*Cm^R^*, *upp pyrR**, *ldh substitution cassette for sadh hydG for C. acetobutylicum*	This study
	*Ap^R^*, *MLS^R^*, *acetone operon*, *repL gene*, *colE1 origin*	
pSOS95	*Ap^R^*, *MLS^R^*, *pyrR*, *repL gene*, *colE1 origin*	[24]
pSOS95*-pyrR*		This study

**Table 2 microorganisms-11-02696-t002:** Oligonucleotides used for PCR amplification.

Primer Name	5′–3′ Oligonucleotide Sequence
PSC 39	GCATGCTCTTGTAGGTGATCCTT
PSC 40	TGTTTACTGAATCCTCTTCATCTATTCC
PSC 46	AAAAAAGGCGCCCTACAACTCATAAATGTTTACTGAATCCTC
PSC 51	CAGAGTATTTAAGCAAAAACATCGTAGAAAT
PSC 52	TTATTTTGTACCGAATAATCTATCTCCAGC
PSC 58	AAAAAAGGATCCTTATACTGGAGGTGAGTGTATGAATTTAAAAG
PSC 61	CCATGGTTATACTGGAGGTGAGTGTATGAATCTTAAAGCTAAGATTCTTGATGATAAGGC
PSC 62	AAACACCGTATTTCTACGATGTTTTTGCTTAAATACTCTGCCATGGCTATAGCTCATATATGTTAACACTATCCTCTTC
PSC 72	TCTTGGAGATGCTGGAGATAGATTATTCGGTACAAAATAACCATGGTTATACTGGAGGTGAGTG
PSC 75	TTAATAGGATCCGAACCCATCAAATAAGAGTGCATATGG
PSC 76	TATTAAGGATCCAGTCCTGCCCAACC
PSC 104	AAATATAAATGAGCACGTTAATCATTTAACATAGATAATTAAATAGTAAAAGGAGGAACATATTTTATGAAAGGTTTTGC
PSC 105	GGCAAAAGTTTTATAAACATGGGTACTGGTTATATTATATTATTTATGACTTTATTATTTATCACCTCTGCAACCACAGC
PSC 106	TAGAGAAATTTTTAAAGATTTCTAAAGGCCTTTAACTTCATGTGAAAAGTTTGTTAAAATATAAATGAGCACGTTAATCATTTAA
PSC 107	TCCACCCTTGGAGTTTAGGTCTTTTACCAGGCCTGAATACCCATGTTTATAGGGCAAAAGTTTTATAAACATGGGTACT
PSB 384	GGGAAAGGTTTTAAGAGCGGCG
PSB 385	CAACAATTGTCTCCGGTTTCAAGGG

**Table 3 microorganisms-11-02696-t003:** Nucleotide sequence of wild-type *pyrR* gene and *pyrR** gene.

Gene Name	Nucleotide Sequence
wild-type *pyrR*	ATGAATTTAAAAGCAAAGATTTTAGATGATAAGGCTATGCAAAGGACTTTGACCAGAATAGCACATGAAATTATAGAAAAGAATAAAGGTATAGATGATATAGTACTAGTAGGAATAAAGAGAAGAGGAGTTCCAATAGCCGATAGAATAGCGGATATAATTGAAGAAATAGAAGGAAGTAAGGTTAAGCTAGGAAAAGTAGATATAACCTTATATAGAGACGATTTGTCTACGGTAAGTTCTCAACCAATAGTAAAAGATGAGGAAGTATATGAAGATGTAAAGGATAAGGTAGTAATACTTGTTGATGACGTTTTATATACAGGAAGAACATGCAGAGCAGCCATAGAAGCTATTATGCATAGAGGAAGACCAAAGATGATACAGCTTGCAGTTTTGATAGATAGGGGACATAGAGAACTTCCTATAAGGGCAGATTATGTTGGAAAAAATGTACCTACATCAAAAAGTGAATTGATATCGGTAAATGTTAAAGGAATAGATGAAGAGGATTCAGTAAACATTTATGAGTTGTAG
synthetic *pyrR (pyrR*)*	ATGAATCTTAAAGCTAAGATTCTTGATGATAAGGCAATGCAAAGGACACTAACCAGAATAGCTCATGAAATAATAGAAAAGAATAAAGGAATAGATGATATAGTTTTGGTTGGAATAAAGAGAAGAGGAGTACCTATAGCGGATAGAATAGCCGATATAATAGAAGAAATAGAAGGATCAAAGGTAAAGTTGGGAAAAGTTGATATAACCCTTTATAGAGACGATCTATCAACCGTTTCAAGTCAACCTATAGTTAAAGATGAGGAAGTTTATGAAGATGTTAAGGATAAGGTTGTTATATTAGTTGATGACGTACTTTATACTGGAAGAACTTGCAGAGCTGCGATAGAAGCAATAATGCATAGAGGAAGACCTAAGATGATACAGTTAGCTGTACTAATAGATAGGGGACATAGAGAACTACCAATAAGGGCTGATTATGTAGGAAAAAATGTTCCAACTAGTAAATCAGAATTGATATCCGTAAATGTAAAAGGAATAGATGAAGAGGATAGTGTTAACATATATGAGCTATAG

**Table 4 microorganisms-11-02696-t004:** Primers used for locus verification after metabolic engineering.

Primers Name	Function
PSC 39–PSC 40	*pyrR* gene sequencing
PSB 384–PSB 385	*ldh locus* verification

**Table 5 microorganisms-11-02696-t005:** Comparison of the bactericidal effect of 5FU, between rich (CGM MES) and synthetic (SM MES) media, at different concentrations, on *C. acetobutylicum ∆cac1502* strain.

5FU Concentration (µM)	CGM MES (UFC/mL)	SM MES (UFC/mL)
0	2.06 ± 0.41 × 10^7^	1.75 ± 0.25 × 10^7^
5	1.65 ± 0.38 × 10^7^	0
25	3.02 ± 0.65 × 10^6^	0
50	1.45 ± 0.23 × 10^6^	0
100	0	0
200	0	0

**Table 6 microorganisms-11-02696-t006:** Protective effect of uracil against 5FU in *C. acetobutylicum ∆cac1502* strain. 0.1 mL of a 10^−1^ dilution of a CGM culture was spread on synthetic medium (SM MES plates) containing 5 µM of 5FU and different uracil concentrations.

**Uracil Concentration (µM)**	**Number of Colonies (5 µM 5FU)**
0	0
5	Layer
12.5	Layer
25	Layer
50	Layer

**Table 7 microorganisms-11-02696-t007:** Spontaneous mutations found in PyrR after exposition of *C. acetobutylicum* strain *Δcac1502* to high 5FU concentrations on synthetic medium (SM MES).

5FU Concentration (µM)	Amino Acid Change	Nucleotide Change
25	R124G	*g.370A>G*
25	A47D	*g.140C>A*
50	R136X	T addition in aa 132
50	P45L	*g.134C>T*
50	V85X	G deletion in aa 85
50	E23K	*g.67G>T*

**Table 8 microorganisms-11-02696-t008:** Evaluation of the *pyrR*^mut^ strain viability on 5FU at different concentrations while maintaining a pCat-*upp* (+Tm).

	**pCat-** ** *upp* ** **-** ** *Δ* ** ** *ldh* **		
5FU concentration (µM)	0	25	50
SM MES (UFC/mL)	2.03 ± 1.19 × 10^7^	8.08 ± 1.72 × 10^6^	7.60 ± 3.60 × 10^6^
SM MES + Tm (UFC/mL)	1.59 ± 0.92 × 10^7^	5.35 ± 3.96 × 10^6^	2.84 ± 1.44 × 10^6^

**Table 9 microorganisms-11-02696-t009:** Evaluation of the *pyrR*^mut^ strain viability, containing a pCat-*upp* plasmid and a replicative pSOS95-*pyrR* plasmid, on 5FU at different concentrations.

	**pCat-** ** *upp* ** **-** ***Δldh* + pSOS95-** ** *pyrR* **			
5FU concentration (µM)	0	5	10	25
SM MES + Ery (UFC/mL)	4.91 ± 1.99 × 10^6^	3.60 ± 2.40 × 10^4^	3.87 ± 2.74 × 10^4^	2.02 ± 2.80 × 10^3^
SM MES + Ery + Tm (UFC/mL)	2.85 ± 1.66 × 10^6^	0	0	0

**Table 10 microorganisms-11-02696-t010:** Evaluation of the *pyrR^mut^* strain containing a pCat-*upp* plasmid and a replicative pSOS95-*pyrR* plasmid for both viability (% of viable clones) and frequency of double crossing-over (% of Tm sensitive clones) after 5FU selections at different concentrations.

	**pCat-** ** *upp* ** **-** ***Δldh* + pSOS95-** ** *pyrR* **		
5FU concentration (µM)	5	10	25
Picked colonies viability (%)	86	90	84
Picked colonies Tm sensitivity (%)	86	96	98

**Table 11 microorganisms-11-02696-t011:** Evaluation of the *pyrR*^mut^ strain viability on 5FU at different concentrations while maintaining or not (+/− Tm) a pCat-*upp-pyrR** plasmid.

	**pCat-** ** *upp* ** **-** ** *pyrR** ** **-** ** *Δldh* **		
5FU concentration (µM)	0	25	50
SM MES (UFC/mL)	3.42 ± 2.42 × 10^7^	3.87 ± 0.37 × 10^4^	4.19 ± 0.31 × 10^4^
SM MES + Tm (UFC/mL)	2.28 ± 1.28 × 10^7^	0	0

**Table 12 microorganisms-11-02696-t012:** Comparison of both viability (% of viable clones) and frequency of double crossing-over (% of Tm sensitive clones) between a *pyrR^mut^* strain containing a pCat-*upp* or a pCat*-upp-pyrR** plasmid after 5FU selections at different concentrations.

	**pCat-** ** *upp* ** **-** ** *Δldh* **				**pCat-** ** *upp* ** **-** ** *pyrR** ** **-** ** *Δldh* **			
5FU concentration (µM)	5	10	25	50	5	10	25	50
Picked colonies viability (%)	100	98	98	76	88	96	60	52
Picked colonies Thiamphenicol sensitivity (%)	0	6	75	89	90	98	100	100

**Table 13 microorganisms-11-02696-t013:** Evaluation of *Δcac1502ΔuppΔcac3535* strain containing a pCat-*upp-pyrR** plasmid for both viability (% of viable clones) and frequency of double crossing over (% of Tm sensitive clones) after 5FU selections at different concentrations.

	**pCat-** ** *upp* ** **-** ** *pyrR** ** **-** ** *Δldh* **	
5FU concentration (µM)	5	10
Picked colonies viability (%)	98	100
Picked colonies Thiamphenicol sensitivity (%)	100	100

**Table 14 microorganisms-11-02696-t014:** Evaluation of the *pyrR^mut^* strain containing the pCat-*upp-pyrR*-Δldh::sadh-hydG* plasmid for both viability (% of viable clones) and frequency of double crossing over (% of Tm sensitive clones) after 5FU selection at 5 µM.

	**pCat** ** *-upp-pyrR*-Δldh::sadh-hydG* **
5FU concentration (µM)	5
Picked colonies viability (%)	92
Picked colonies’ thiamphenicol sensitivity (%)	100

## Data Availability

The nucleotide sequence of the *CAB1060* strain will be provided upon request to the corresponding author.

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
