# Peer review of "Improvement of the Genome Editing Tools Based on 5FC/5FU Counter Selection in Clostridium acetobutylicum"

_microorganisms, 2023, doi:10.3390/microorganisms11112696_

Round 1
Reviewer 1 Report
Comments and Suggestions for Authors
1. The improved effeminacy of the method utilized here should be addressed. The problems and measures in the experiment and the comparison with other methods should be discussed.
2. What is the merits of the 5FC/5FU selection? The novelty should be described in introduction portion.
3. The detailed information on materials and methods should be improved. The information of the kit, instruments and software should be provided, including Model, manufacturer, country. The PCR program and amplification system should be added. Some data such as Table 3should move to Supplementary material, and there is a mistake in the writing of the molecular formula of chemical reagents
4. What's the Viability and frequency? How did you get it?
5. How about the genetic and functional stability of mutants obtained by the genome editing tools based on 5FC/5FU 2 counter selection。
6. The discussion portion seems miss the focus. The problems and measures in the experiment and the comparison with other methods should be added to the discussion.
Comments on the Quality of English LanguageNone
Reviewer 2 Report
Comments and Suggestions for Authors
In this study, after multiple applications of the pCat-upp/5FU system for genome modification in Clostridium acetobutylicum, the bacteria will become completely resistant to 5FU in the presence of the upp gene. Two plasmids were developed, one overexpressing the native pyrR gene, another suicide plasmid carrying the unmutated and optimized pyrR gene (pyrR*) and upp, restored the 5FU sensitivity of the strain and reduced the 5FU concentration from 1mM to 5µM by using a defined synthetic medium, and after validating a new genome editing method for Clostridium acetobutylicum, inserting sadh and hydG of Clostridium beijerinckii at the ldh site to produce isopropanol. However, there are still several points that need to be revised and clarified to support the authors's study.
1. In section 2.4, the authors need to explain what closed frequency codons are.
2. In section 2.8, the authors should explain what operator proteins the sadh and hydG genes encode respectively.
3. In section 3.1, please supplement the genome sequencing results of the strain and list all random mutation sites.
4. In section 3.2, the authors need to explain why the addition of yeast extract results in a decrease in the protective effect of uracil.
5. In section 3.3, please explain how to judge that the upp gene contained in the suicide plasmid is sufficient to obtain the sensitivity of the strain to 5FU.
6. In section 3.5, please evaluate the viability of C. acetobutylicum strains with pCat-upp-pyrR*-Δldh integrated into the ldh locus plasmid at higher 5FU concentrations.
7. Please use the same format as the previous table for Table 7-13.
8. In figure 5 of part 4, the authors please give an overall summary of the logic of the article.
Comments on the Quality of English LanguageI have no comments on the quality of english language.
Reviewer 3 Report
Comments and Suggestions for Authors
Dear Authors,
Please see my Review Report in the attached file.
Thank You & Best Regards!

The manuscript would benefit from a moderate linguistic refinement.
Round 2
Reviewer 1 Report
Comments and Suggestions for Authors
The author has revised all that I concerned.
Reviewer 3 Report
Comments and Suggestions for Authors
Dear Authors,
Upon re-evaluation of the manuscript following its major revision, it is evident that the authors have taken significant steps to address the initial concerns and feedback. The revisions have led to a substantial enhancement in the quality, clarity, and depth of the work presented. The current version meets the rigorous standards expected for publication in the Microorganisms Journal.
I would, therefore, recommend this manuscript for publication.
Congratulations to the authors on their dedicated effort to improve and refine their manuscript.
Best Regards!